# Leave No One Behind: Online Self-Supervised Self-distillation for Sequential Recommendation

## ABSTRACT

Sequential recommendation methods play a pivotal role in modern recommendation systems. A key challenge lies in accurately modeling user preferences in the face of data sparsity. To tackle this challenge, recent methods leverage contrastive learning (CL) to derive self-supervision signals by maximizing the mutual information of two augmented views of the original user behavior sequence. Despite their effectiveness, CL-based methods encounter a limitation in fully exploiting self-supervision signals for users with limited behavior data, as users with extensive behaviors naturally offer more information. To address this problem, we introduce a novel learning paradigm, named Online Self-Supervised Self-distillation for Sequential Recommendation ($S^4$Rec), effectively bridging the gap between self-supervised learning and self-distillation methods. Specifically, we employ online clustering to proficiently group users by their distinct latent intents. Additionally, an adversarial learning strategy is utilized to ensure that the clustering procedure is not affected by the behavior length factor. Subsequently, we employ self-distillation to facilitate the transfer of knowledge from users with extensive behaviors (teachers) to users with limited behaviors (students). Experiments conducted on four real-world datasets validate the effectiveness of the proposed method.

## CCS CONCEPTS

• **Information systems** → **Recommender systems**; • **Computing methodologies** → **Learning paradigms**.

## KEYWORDS

Sequential Recommendation, Multi-Intention Modelling, Self-Supervised Learning, Long-Tail Learning

**ACM Reference Format:**
Anonymous Author(s). 2018. Leave No One Behind: Online Self-Supervised Self-distillation for Sequential Recommendation. In *Proceedings of Make sure to enter the correct conference title from your rights confirmation emai (Conference acronym 'XX).* ACM, New York, NY, USA, 9 pages. https://doi.org/XXXXXXX.XXXXXXX

## 1 INTRODUCTION

As an important recommendation paradigm, sequential recommendation has been playing a vital role in online platforms, e.g., Amazon and Alibaba. Generally, sequential recommendation takes a

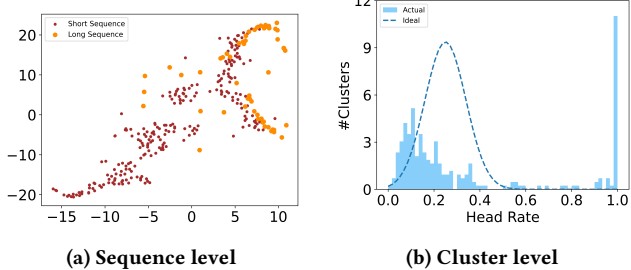

**(a) Sequence level**    **(b) Cluster level**

**Figure 1: Visualization of clustering for sequence granularity and cluster granularity on an amazon dataset.**

sequence of user-item interactions as the input and aims to predict the subsequent user-item interactions that may happen in the near future through modelling the complex sequential dependencies embedded in the sequence of historical interactions. Early works based on Markov Chains [10, 23] focus on modelling simple low-order sequential dependencies. Afterward, deep learning networks, such as recurrent neural networks (RNN) [11, 12], convolutional neural networks (CNN) [26, 34], and memory networks [14] have drawn attention for sequential recommendations due to the powerful non-linear expressive capacity. In addition, transformer-based [15, 24, 28] models have gained popularity for sequential recommendations. They can effectively learn users' preferences by estimating an importance weight for each item.

Although these methods have achieved promising results, they usually only utilize the item prediction task to optimize a huge amounts of parameters, which suffers from data sparsity problem easily. To tackle the problem, inspired by the successes of self-supervised learning in computer vision (CV) [3] and natural language processing (NLP) [8], recent works attempt to use self-supervised learning techniques to optimize the user representation model for improving sequential recommendation systems. These methods typically derive self-supervision signals through maximizing the mutual information of two augmented views of the original user behavior sequence.

Despite their effectiveness, aforementioned methods fail to further extract supervision information across historical interactions. In practice, users consume each item based on their latent intents, which can be perceived as a subjective motive for their interaction. This motivates the exploration [19, 21] to extract shared underlying intents among users, which can be utilized to guide the recommendation system in providing more relevant recommendations. Since these methods require labels to model the user's intents, ICLRec [5] learns users' underlying intent distributions from all user interaction sequences via clustering. However, clustering algorithms

typically involve operations over entire datasets, which can be computationally challenging and less efficient dealing with large-scale datasets.

Furthermore, these methods also encounter a limitation in fully exploiting self-supervision signals for users with limited behavior data, as users with extensive behaviors naturally offer more information. As illustrated in Figure 1, the learned representations of users with extensive behaviors (long sequences) tend to be clustered by themselves which are relatively separated from users with limited behaviors (short sequences). However, the learned user representations should be affected by users' latent intents and unaffected by the observed sparsity of behavior sequence. Many studies point that uniform representation distribution is a crucial factor for the performance of contrastive learning methods [30, 33]. Previous CL-based and intention modelling methods fail to handle the distribution discrepancy between these two types of users, which hinders the sequence recommendation performance, especially for the users with limited behaviors.

To address these problems, we introduce a novel learning paradigm, named Online Self-Supervised Self-distillation for Sequential Recommendation ($S^4$Rec), effectively bridging the gap between self-supervised learning and self-distillation methods. Specifically, we employ online clustering to proficiently group users by their distinct latent intents. Additionally, an adversarial learning strategy is utilized to ensure that the clustering procedure is not affected by the behavior length factor. Subsequently, we employ self-distillation to facilitate the transfer of knowledge from users with extensive behaviors (teachers) to users with limited behaviors (students).

The main contributions of this paper are summarized as follows:

- We propose a novel learning paradigm for sequential recommendation, which bridges the gap between self-supervised learning and self-distillation methods. To the best of our knowledge, this is the first work to apply self-distillation techniques to the sequential recommendation.
- We propose online clustering and adversarial learning modules to learn user representation clusters which are unaffected by the sparsity of behavior. Based on the learned clusters, the cluster-aware self-distillation module is employed to transfer knowledge from users with extensive behaviors to users with limited behaviors.
- Extensive experiments are conducted on four real-world datasets, which show the state-of-the-art performance of the proposed $S^4$Rec model.

## 2 RELATED WORK

### 2.1 Sequential Recommendation

Sequential recommendation aims to learn users' interests and forecast the next items they would most like to interact with by modeling the sequences of their historical interactions.

Early works based on Markov Chains [10, 23] focus on modeling simple low-order sequential dependencies. These approaches rely on rigorous assumptions and are powerless to handle complex patterns. Afterward, deep learning networks, such as recurrent neural networks (RNN)[11, 12], convolutional neural networks (CNN)[26, 34] and memory networks [14] have drawn attention for

sequential recommendations due to the powerful nonlinear expressive capacity. Recently, transformer-based [28] models have gained popularity for sequential recommendations. Typically, SASRec [15] uses self-attention mechanism to dynamically assign weights to each item. BERT4Rec [24] proposes a deep bidirectional transformer model to extract both left and right-side behaviors information. ASReP [20] further solves data sparsity problem by introducing a pretrained transformer on the revised interaction sequences to augment short sequences.

### 2.2 Intention Learning for Recommendation

Many approaches have been proposed to study users' intents behind each user's behavior for improving recommendations [4, 21, 25].

DSSRec [21] introduces a sequence2sequence training strategy to capture extra supervision information. An intent variable is employed to extract mutual information between an individual user's past and future interaction sequences. ICLRec [5] learns users' intent distribution from unlabeled user behavior sequences and optimize SR models with contrastive learning by considering the obtained intents. ISRec [17] extracts the intentions of the target user from sequential contexts, then takes complex intent transition into account through the message-passing mechanism on an intention graph.

### 2.3 Self-Supervised Learning for Recommendation

Self-Supervised Learning (SSL) become prevalent in different research areas, including computer vision [3], natural language processing [8], and more. The main target of SSL is to capture high-quality and information-rich representations through the feature itself. There have also been some recent works to apply SSL to sequential recommendations. For example, $S^3$-Rec [36] adopts a pretraining and fine-tuning strategy with four self-supervised tasks, and first proposes to maximize the mutual information between historical items and their attributes. CL4SRec [31] introduces three data-level augmentation approaches (crop/mask/reorder, referred to as invasive augmentation methods in the paper) to structure positive views. Later, CoSeRec [19] aims to produce robust augmented sequences based on item denpendencies since random item perturbations may weaken the confidence of positive pairs. ICLRec [5] conducts clustering among all user behavior sequences to obtain user's intent, and optimizes sequential recommendation model by maximizing the mutual information between sequence and corresponding intentions.

## 3 PRELIMINARIES

### 3.1 Problem Formulation

We denote $\mathcal{U}$ and $\mathcal{I}$ as the user set and item set, respectively. For each user $u \in \mathcal{U}$, his/her chronological interaction sequence can be represented as $\mathcal{S}_u = [s_u^1, ..., s_u^l, ..., s_u^L]$, where $s_u^l$ denotes the $l$-th item that user $u$ interacted and $L$ is the maximum sequence length. The goal of sequential recommendation is to predict the next item $s_u^{L+1}$ which the user $u$ will most likely interact with given the behavior sequence $\mathcal{S}_u$. To this end, the classical objective function

for SR is usually formalized as follows:

$$\mathcal{L}_{SR} = \sum_{u=1}^{|\mathcal{U}|} \sum_{l=2}^{|L|} -\log p_\theta(s_u^{l+1}|s_u^1, s_u^2, ..., s_u^l),\qquad(1)$$

where $\theta$ is the parameters of a neural network $f_\theta$ that encodes sequential feature into latent vectors: $\mathbf{z}_u = f_\theta(\mathcal{S}_u)$. The probability $p(s_u^{l+1}|\mathbf{z}_u^l)$ is computed based on the similarity between the encoded sequential patterns $\mathbf{z}_u^l$ and the representation of the next item $s_u^{l+1}$. In serving stage, the items with the highest probability will be recommended to the user $u$.

## 3.2 Sequence Augmentation Operators

Given an original behavior sequence $\mathcal{S}_u$, several random sequence-level augmentation strategies can be employed [19, 31]:

- **Mask.** It randomly masks a proportion of items in an original sequence. This mask operation can be formulated as:

$$\mathcal{S}_u^{Mask} = [\hat{s}_u^1, ..., \hat{s}_u^l, ..., \hat{s}_u^L],\qquad(2)$$

where $\hat{s}_u^l$ represents the masked item if $s_u^l$ is selected, otherwise $\hat{s}_u^l = s_u^l$.
- **Crop.** It randomly removes a continuous sub-sequence from positions $l$ to $l + l_c$ in $\mathcal{S}_u$. The length to crop is set by $l_c = \delta * |\mathcal{S}_u|$ where empirically $\delta = 0.8$. The formulation of the cropped sequence is shown below:

$$\mathcal{S}_u^{Crop} = [s_u^1, ..., s_u^l, s_u^{l+l_c}, ...., s_u^L],\qquad(3)$$

- **Reorder.** It randomly shuffles a continuous sub-sequence from positions $l$ to $l + l_c$ in $\mathcal{S}_u$. The length to reorder is set by $l_c = \delta * |\mathcal{S}_u|$ where empirically $\delta = 0.2$. The formulation of the reordered sequence is as:

$$\mathcal{S}_u^{Reorder} = [s_u^1, ..., \hat{s}_u^l, ..., \hat{s}_u^{l+l_c}, ...., s_u^L],\qquad(4)$$

- **Insert.** It inserts an item chosen randomly from the interaction histories of other users into a randomly selected position within $\mathcal{S}_u$. This operation is employed repeatedly on the sequence to obtain an augmented view. The augmented sequence could be formulated by:

$$\mathcal{S}_u^{Insert} = [s_u^1, ..., \hat{s}_u^1, ..., \hat{s}_u^i, ...., s_u^L].\qquad(5)$$

## 3.3 Latent Intent Modeling in SR

Due to subjective reasons, while users face various items in a recommendation system, they may have multiple intentions (e.g., purchasing outdoor equipment, preparing for lectures, just killing time, etc.). The intent variable can be formed as $\mu \in \mathbb{R}^{K \times d}$. Then the probability of a user interacting with a certain item can be rewritten as $\mathbb{E}_\mu[p(s_u^{l+1}|\mathbf{z}_u^l, \mu)]$. As users intents are usually implicit, some work [5] attempts to infer this latent intents by unsupervised approach, such as clustering.

## 4 METHODOLOGY

In this section, we discuss the details of our proposed $S^4$Rec. The overall framework is illustrated in Figure 2.

## 4.1 Clustering On The Fly

Previous work learns users' implicit intents based on user interaction data typically employ clustering methods, such as ICLRec[5]. It firstly encodes all the sequences $\{\mathcal{S}_u\}_{u=1}^{|\mathcal{U}|}$ by a sequence encoder $f_\theta$. Subsequently, ICLRec executes $K$-means clustering over all the sequence representations $\{\mathbf{z}_u\}_{u=1}^{|\mathcal{U}|}$ to obtain cluster assignment $\mathbf{P} \in \mathbb{R}^{|\mathcal{U}| \times K}$.

However, one main issue of these clustering-based methods is that they do not scale well with the dataset as they require a pass over the entire dataset to capture cluster assignments that are used as targets during training. In addition, there is no correspondence between two consecutive cluster assignments. Hence, the final prediction layer learned from an assignment may become irrelevant for the following one and thus needs to be reinitialized from scratch at each epoch, which considerably disrupts the model training. In this work, we describe an alternative [2] to mapping sequence representations to prototype latent space on the fly in order to scale to large uncurated datasets, and thus retain correspondence.

Firstly, the original interaction sequence is mapped into a vector representation by an encoder as following:

$$\mathbf{z}_u = f_\theta(\mathcal{S}_u),\qquad(6)$$

where $f_\theta$ is an alternative sequence encoder, which is set as SASRec [15] in this paper.

Then the soft cluster assignment $\mathbf{p}_u$ of $u$ can be calculated as :

$$\mathbf{p}_u^k = \frac{\exp(\mathbf{z}_u \boldsymbol{\mu}_k^\top / \tau_1)}{\sum_{k'} \exp(\mathbf{z}_u \boldsymbol{\mu}_{k'}^\top / \tau_1)},\qquad(7)$$

where $\boldsymbol{\mu}_k$ is the $k$-th row of $\boldsymbol{\mu} \in \mathbb{R}^{K \times d}$, which represents $K$ trainable prototypes, i.e. intent representations. $\tau_1$ is a temperature parameter.

We then further refine the cluster assignment with the help of an auxiliary target distribution $\mathbf{q}_u$, obtained by mapping $\mathbf{z}_u$ to $\boldsymbol{\mu}$. The objective is trained by matching the soft assignment to the target distribution, specifically defined as a cross entropy loss between the soft assignment $\mathbf{p}_u$ and the auxiliary target $\mathbf{q}_u$, i.e.,

$$\mathcal{L}_{Clust}(\mathbf{z}_u, \mathbf{q}_u) = -\sum_k \mathbf{q}_u^k \log \mathbf{p}_u^k.\qquad(8)$$

The objective function is jointly minimized with respect to the prototypes $\boldsymbol{\mu}$ and the parameters $\theta$ of the sequence encoder $f_\theta$ used to produce the sequence representation $\mathbf{z}_u$.

We now introduce the method to obtain the auxiliary target and update the prototypes. To make our proposal cluster online, we iteratively compute the auxiliary target using only the sequence features within a batch. We utilize the prototypes $\boldsymbol{\mu}$ to compute the auxiliary target and enforce all the instances in a batch equally partitioned by the prototypes as much as possible. This equipartition constraint ensures that the auxiliary targets for different sequences in a batch are distinct, thus preventing the trivial solution where every sequence has the same auxiliary target [2].

Given $B$ embedding vectors $\mathbf{Z} = [\mathbf{z}_1, ..., \mathbf{z}_B]$ in a mini-batch, we are interested in mapping them to the prototypes $\boldsymbol{\mu} = [\boldsymbol{\mu}_1, ..., \boldsymbol{\mu}_K]$. We denote this mapping or target distribution as $\mathbf{Q} = [\mathbf{q}_1, ..., \mathbf{q}_B]$,

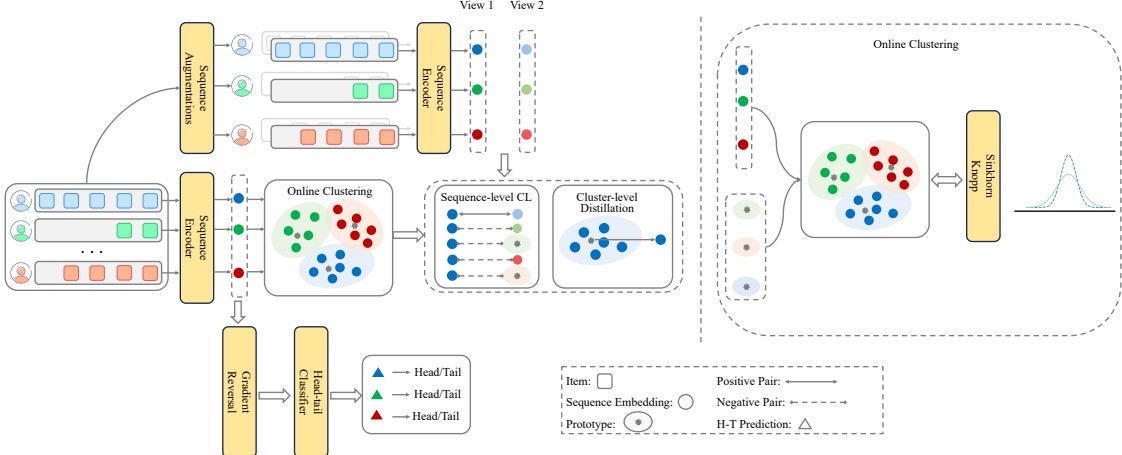

**Figure 2: The overall framework of $S^4$Rec. It augments original behavior sequences as contrastive views and employs online clustering to proficiently group users by their distinct latent intents. Subsequently, $S^4$Rec conducts cluster-aware self-distillation to transfer knowledge from corresponding intents (teachers) that contain extensive behavior information to users with limited behaviors (students). Additionally, an adversarial learning strategy ensures that the the clustering procedure is not affected by the head-tail factor. The right part describes the online clustering process based on the Sinkhorn-Knopp algorithm.**

and optimize $\mathbf{Q}$ to maximize the similarity between sequence embeddings and prototypes:

$$\max_{\mathbf{Q} \in Q} Tr(\mathbf{Q}\boldsymbol{\mu}\mathbf{Z}^\top) + \epsilon H(\mathbf{Q}),$$
$$H(\mathbf{Q}) = -\sum_{ij} \mathbf{Q}_{ij} \log \mathbf{Q}_{ij}, \qquad (9)$$

where $H$ is a entropy function and $\epsilon$ is a parameter adjusting the smoothness of the auxiliary target.

We adopt the solution in [1, 2] that achieves an equal partition by modelling the $\mathbf{Q}$ to belong to the transportation polytope within mini-batch:

$$\mathbf{Q} = \{\mathbf{Q} \in \mathbb{R}_+^{B \times K} | \mathbf{Q}\mathbf{1}_K = \frac{1}{B}\mathbf{1}_B, \mathbf{Q}^\top\mathbf{1}_B = \frac{1}{K}\mathbf{1}_K\}, \qquad (10)$$

where $\mathbf{1}_B$ denotes the vector of ones in dimension of batch size $B$. These constraints enforce that on average each prototype is selected at least $\frac{B}{K}$ times within a mini-batch.

One solution $\mathbf{Q}^*$ of Eq. (9) over the set $Q$ takes the form of a normalized exponential matrix and is as following [6]:

$$\mathbf{Q}^* = diag(\mathbf{m}) \exp(\frac{\mathbf{Z}\boldsymbol{\mu}^\top}{\epsilon}) diag(\mathbf{v}). \qquad (11)$$

where $\mathbf{m} \in \mathbb{R}^B$ and $\mathbf{v} \in \mathbb{R}^K$ are re-normalized vectors that are computed using the iterative Sinkhorn-Knopp algorithm [6], which requires only a small number of matrix multiplications.

## 4.2 Cluster-aware Self-distillation

Once the prototypes $\boldsymbol{\mu}$ and corresponding clustering assignments $\mathbf{p}_u$ are obtained, they are employed to construct the supervisory signals for the self-supervision task. More precisely, we propose cluster-aware two-fold self-distillation (CSD) modules: a sequence-level contrastive module and a cluster-level self-distillation module. Concretely, the sequence-level contrastive module maximizes

mutual information among the positive augmentation pair of the sequence itself while promoting discrimination ability to the negatives. In parallel, the cluster-level self-distillation module aligns each user's behavior sequence to its corresponding intents consistently. The detail is described as follow.

*4.2.1 Sequence-level Contrastive Learning.* Users' sequential behaviors naturally present extensive information for obtaining self-supervised signals. Given a function set $\mathcal{G}$ of several data augmented operators, such as **mask**, **crop**, **reorder** and **insert** [18] and an user's sequence $\mathcal{S}_u$, we can create two augmented views as:

$$\tilde{\mathcal{S}}_u^1 = g_u^1(\mathcal{S}_u), \tilde{\mathcal{S}}_u^2 = g_u^2(\mathcal{S}_u), g_u^1, g_u^2 \in \mathcal{G}, \qquad (12)$$

where $g_u^1$ and $g_u^2$ are augmentation functions sampled from $\mathcal{G}$ to produce different views for $\mathcal{S}_u$. Generally, views captured from the same sequence are treated as positive pairs, while those of any other sequences are considered as negative pairs. Furthermore, the contrastive views are mapping into representations $\mathbf{z}_u^1$ and $\mathbf{z}_u^2$ by the sequence encoder $f_\theta$. After that, we can maximize the mutual information to provide self-supervised signals to improve recommendation performance:

$$\mathcal{L}_{SCL} = \mathcal{L}_{SCL}(\mathbf{z}_u^1, \mathbf{z}_u^2) + \mathcal{L}_{SCL}(\mathbf{z}_u^2, \mathbf{z}_u^1),$$
$$\mathcal{L}_{SCL}(\mathbf{z}_u^1, \mathbf{z}_u^2) = -\log \frac{\exp(\mathbf{z}_u^1\mathbf{z}_u^{2\top}/\tau_2)}{\sum_{n \neq u} \exp(\mathbf{z}_u^1\mathbf{z}_n^\top/\tau_2)}, \qquad (13)$$

where $\mathbf{z}_n$ are negative views' representations of sequence $\mathcal{S}_u$ and $\tau_2$ is the temperature parameter tuning the strength of penalties on the hard negative samples [29].

The classical InfoNCE [27] loss Eq. (13) only utilized the supervision information of the sample itself. Now that we have gained users' intent, we consider further pushing users' embedding away from the cluster they do not belong to by adding extra negative

intent information. Thus $\mathcal{L}_{SCL}(\cdot, \cdot)$ can be rewritten as:

$$\mathcal{L}_{SCL}(\mathbf{z}_u^1, \mathbf{z}_u^2) =$$
$$-\log \frac{\exp(\mathbf{z}_u^1 \mathbf{z}_u^{2\top}/\tau_2)}{\sum_{n \neq u} \exp(\mathbf{z}_u^1 \mathbf{z}_n^\top/\tau_2) + \sum_{k' \neq h(u)} \exp(\mathbf{z}_u^1 \boldsymbol{\mu}_{k'}^\top/\tau_2)}, \quad (14)$$

where $h(u)$ is a function maps $u$ into the index of assigned intent.

*4.2.2 Cluster-level Distillation Module.* Considering the sparsity of most users' sequences, the capability of supervisory signals within the instance scope is limited. Knowledge Distillation (KD) [13] is a preferred choice for alleviating sparsity problems. However, most KD methods require a large pretrained, and relatively complex teacher model, which cannot be met in all cases.

Confronting the expensive KD process, self-distillation [32, 35] is proposed to eliminate the requirement of complex teacher model by sharing the same backbone network to serve as teachers and students simultaneously. Hence, the distillation process is significantly simplified. Nevertheless, this work distinguishes teachers and students through distinct sub-network configurations, which is not preferred due to the complexities.

We propose a cluster-aware self-distillation loss that encourages the sequence embedding (student) to be close to the assigned intent distribution (teacher). Remember that the assigned intent of $u$ is denoted as $\boldsymbol{\mu}_{h(u)}$. First, we derive the normalized distribution for sequence embedding and corresponding intent embedding:

$$\mathbf{e}_{u,t}^i = \frac{\exp(\boldsymbol{\mu}_{h(u)}^i/\tau_3)}{\sum_{i'} \exp(\boldsymbol{\mu}_{h(u)}^{i'}/\tau_3)},$$
$$\mathbf{e}_{u,s}^i = \frac{\exp(\mathbf{z}_u^i/\tau_3)}{\sum_{i'} \exp(\mathbf{z}_u^{i'}/\tau_3)}, \quad (15)$$

Here $\tau_3$ is a hyper-parameter called distillation temperature, which controls the smoothness of the normalized distribution. Then the cluster-aware self-distillation loss can be directly defined as KL divergence:

$$\mathcal{L}_{CKD} = \mathcal{L}_{CKD}(\boldsymbol{\mu}_{h(u)}, \mathbf{z}_u^1) + \mathcal{L}_{CKD}(\boldsymbol{\mu}_{h(u)}, \mathbf{z}_u^2),$$
$$\mathcal{L}_{CKD}(\boldsymbol{\mu}_{h(u)}, \mathbf{z}_u^1) = \sum_i \mathbf{e}_{u,t}^i \log \frac{\mathbf{e}_{u,t}^i}{\mathbf{e}_{u,s}^i}, \quad (16)$$

The distillation loss makes use of intent representation which provides additional supervision signals to the sequence embedding and endows the generalization ability to infer the next item.

## 4.3 Head-tail Adversarial Learning

Clustering can easily group long and short sequences into separate clusters, indicating that the clusters possess semantic information regarding the sequence sparsity. While the tail sequences are clustered together, the information beyond the length of the sequence within the cluster would be very sparse, thereby impacting the efficacy of distillation on the tail sequences.

Taking inspiration from the advancement in generative models [9], we propose incorporating an additional adversarial task of head-tail classification. This new task aims to promote a more uniform distribution of head and tail sequences across different clusters, ultimately boosting the overall recommendation performance.

To achieve this, we enhance the recommendation model by introducing a classifier that utilizes the learned sequence embeddings as input. The objective is to train the classifier to accurately predict the category of each sequence. Simultaneously, we aim to optimize the encoder to generate sequence embeddings that can effectively deceive the classifier. By jointly training the classifier and the encoder, we strive to eliminate the length information, leading to a more uniform clustering distribution.

In our experiments, we employ a fully connected layer as the classifier for sequence category classification and utilize cross entropy loss to optimize the classifier. The formula is:

$$\hat{\mathbf{c}} = \mathbf{W}\mathbf{z}_u,$$
$$\mathcal{L}_{Adv} = -\hat{\mathbf{c}}[c] + \log(\sum_i \exp(\hat{\mathbf{c}}[i])), \quad (17)$$

where $\hat{\mathbf{c}}$ is output logits of the classifier, and $c$ is the corresponding category of sequence $\mathcal{S}_u$. Under the setting of adversarial learning, the object for the sequence category classifier is to minimize $\mathcal{L}_{Adv}$, and the object for the encoder is to minimize $\mathcal{L}_{SR} - \gamma \mathcal{L}_{Adv}$, where $\gamma$ is introduced to balance the main task and the additional adversarial task.

With respect to the classifier, the classification loss is minimized by finding the category of sequence embeddings. While for the recommendation model, the classification loss is reversed which pushes sequence embeddings of the same category far from each other and not to form clusters. Meanwhile, the main task of minimizing the recommendation loss forces the learned embedding space to retain interest preference semantics.

In the context of implementing adversarial learning, one elegant approach is to incorporate a Gradient Reversal Layer (GRL) within the backward propagation process which is initially introduced in DAN [7]. Since we expect the classifier to minimize $\mathcal{L}_{Adv}$, while forcing the main encoder to maximize $\mathcal{L}_{Adv}$, we insert a GRL layer between the main encoder and the fully connected classifier. During the backpropagation process, the gradients for minimizing the classification loss flow backward through the classifier, and after the GRL, the gradients will be reversed, which further flows to the encoder. That is, we perform gradient descent on the parameters of the classifier while performing gradient ascent on the parameters of the encoder, with respect to $\mathcal{L}_{Adv}$. for other objectives, gradient descent is applied to the encoder. Through this subtle design, we successfully implement the adversarial learning task.

With the help of adversarial learning, the impact of the head-tail property sequence would be eliminated to some extent.

## 4.4 Multi-task Training

We adopt a multi-task strategy where the main next-item prediction, the cluster assignment, the cluster-aware self-distillation and the adversarial learning task are jointly optimized. The joint loss is a linear weighted sum calculated as:

$$\mathcal{L} = \mathcal{L}_{SR} + \alpha \mathcal{L}_{Clust} + \beta_1 \mathcal{L}_{SCL} + \beta_2 \mathcal{L}_{CKD} + \lambda \mathcal{L}_{Adv}. \quad (18)$$

where $\alpha$, $\beta_1, \beta_2$ and $\lambda$ are hyper-parameters.

## 5 EXPERIMENTS

### 5.1 Experimental Settings

*5.1.1* ***Datasets.*** We conduct experiments on four widely used benchmark datasets with diverse distributions: **Beauty**, **Sports** and **Toys** are three subcategories constructed from Amazon review datasets [22]; **ML-1M** [1] is a famous movie rating dataset comprising one million ratings. We pre-process these datasets in the same manner following [15, 19, 26] by removing items and users that occur less than five times. Table 1 shows dataset statistics after pre-processing.

*5.1.2* ***Evaluation Metrics.*** Following previous works [5, 19, 24], we adopt two metrics evaluating the performance of SR models: top-$k$ Hit Ratio@$k$ (HR@$k$) and top-$k$ Normalized Discounted Cumulative Gain (NDCG@$k$) with $k$ chosen from $\{5, 20\}$. For each user's behavior sequence, we reserve the last two items for validation and test, respectively, and use the rest to train SR models.

*5.1.3* ***Baseline Models.*** We compare our proposed $S^4$Rec with three categories of methods:

- **Standard sequential models.** Caser [26] is a CNN-based approach, GRU4Rec [12] is an RNN-based method, and SASRec [15] is one of the state-of-the-art Transformer-based baselines for SR. They optimize the same objective but differ in sequence encoder structures.
- **Sequential models considering SSL.** BERT4Rec [24] proposes a deep bidirectional transformer model to extract both left and right-side behaviors information. $S^3$-Rec [36] adopts a pre-training and fine-tuning strategy with four self-supervised tasks. CL4SRec [31] introduces three data-level augmentation approaches to construct positive views. This line of works all utilize the transformer as sequence encoder but adopt distinct contrastive learning tasks.
- **Sequential models with additional latent factors.** DSSRec [21] introduce an intent variable to extract mutual information between an individual user's past and future interaction sequences. ICLRec [5] leverages the clustered latent intent factor and contrastive self-supervised learning to optimize SR.

*5.1.4* ***Implementation Details***. For BPR-MF and GRU4Rec, we use the source code provided by Wang et al.[2] in PyTorch. For Caser, SASRec, BERT4Rec and $S^3$Rec, the source code is provided by Zhao et al.[3] in PyTorch. For DSSRec[4], ICLRec[5] and CL4SRec, we use the source code provided by their authors. Our method is implemented in PyTorch as well. For all models, the dimension of embedding is set as 64, and the maximum sequence length is set as 50 for alignment, following previous works [15, 24, 36]. For each baseline model, all other hyper-parameters are set following the suggestions from the original papers.

For our proposed $S^4$Rec, the optimizer is Adam [16], learning rate is 0.001, batch size $B$ is 512, dropout rate is 0.5, number of clusters $K$ is 128, number of hidden layers is set from $\{1, 2, 3\}$. Multi-task

---

[1]grouplens.org/datasets/movielens/1m
[2]https://github.com/THUwangcy/ReChorus
[3]https://github.com/RUCAIBox/RecBole
[4]https://github.com/abinashsinha330/DSSRec
[5]https://github.com/salesforce/ICLRec

**Table 1: Statistics of used datasets.**

| Datasets | Beauty | Sports | Toys | ML-1M |
|---|---|---|---|---|
| #Users | 22363 | 35598 | 19412 | 6041 |
| #Items | 12101 | 18357 | 11924 | 3417 |
| #Actions | 0.2m | 0.3m | 0.17m | 0.99m |
| Avg.length | 8.9 | 8.3 | 8.6 | 165.5 |
| Sparsity | 99.95% | 99.95% | 99.93% | 95.15% |

objective weights $\alpha, \beta_1, \beta_2, \lambda \in \{0, 0.01, 0.1, 1.0\}$. The temperature parameters $\tau_1, \tau_2, \tau_3$ are chosen from $\{0.1, 1.0\}$.

### 5.2 Overall Performance

In Table 2, we present the consistent performance gain of the proposed $S^4$Rec against baselines on different datasets. The major results are summarized as follows:

- $S^4$Rec achieves remarkable improvement over the strongest baseline ICLRec w.r.t HR@5 by 2.63%∼9.26% and NDCG@5 by 2.25%∼10.13%, respectively. It demonstrates that the proposed framework is dataset agnostic and performs stably given distinct behavior distributions. Further, it is beneficial to employ the adversarial classifier to alleviate the distribution discrepancy between head and tail users in the clustering process.
- Compared to standard sequential models, $S^4$Rec indisputably outperforms all benchmarks. Although the Transformer-based encoder achieves the best performance in standard sequential models, it still performs relatively poorly against $S^4$Rec. Thus, it is critical to utilize self-supervised signals to sufficiently optimize model parameters.
- Compared to sequential models with SSL, $S^4$Rec introduces latent prototypes that summarize the semantics of entire user behavior sequences. Although self-supervised signals are utilized in most benchmarks, they focus on augmented behavior-level views to obtain separated user behavior representations but fail to leverage intent-level information from the augmented sequences. As a result, $S^4$Rec achieves considerable performance gains against this line of research.
- The performance of $S^4$Rec leads in each metric compared to ICLRec. The performance of ICLRec is limited as there is no correspondence between two consecutive cluster assignments in ICLRec. Hence, the final prediction layer learned from an assignment may become irrelevant for the following one and thus needs to be reinitialized from scratch at each epoch, which considerably disrupts the model training. On the contrary, $S^4$Rec maps user behavior sequences to K prototypes in an online fashion. In addition, ICLRec suffers from the head-tail problem, which leads to suboptimal clustering. The results validate the effectiveness of the online clustering and adversarial strategy.

### 5.3 Head-tail Study

As discussed in Section 1, the uniform distribution is a significant factor in contrastive learning. Thus, we further study the impact of adversarial learning for head-tail on clustering results through

**Table 2: Overall performance. Bold scores represent the highest results of all methods. Underlined scores stand for the second highest. "∗" denotes the statistical siginificance for $p < 0.01$ compared to the best baseline methods with paired $t$-test.**

| Dataset | Metric | BPR | GRU4Rec | Caser | SASRec | DSSRec | BERT4Rec | $S^3$-Rec | CL4SRec | ICLRec | Ours |
|---|---|---|---|---|---|---|---|---|---|---|---|
| Beauty | HR@5 | 0.0212 | 0.0111 | 0.0251 | 0.0374 | 0.0410 | 0.0360 | 0.0189 | 0.0423 | 0.0475 | **0.0519***|
| | HR@20 | 0.0589 | 0.0478 | 0.0643 | 0.0901 | 0.0914 | 0.0984 | 0.0487 | 0.0994 | 0.1050 | **0.1071*** |
| | NDCG@5 | 0.0130 | 0.0058 | 0.0145 | 0.0241 | 0.0261 | 0.0216 | 0.0115 | 0.0281 | 0.0316 | **0.0348*** |
| | NDCG@20 | 0.0236 | 0.0104 | 0.0298 | 0.0387 | 0.0403 | 0.0391 | 0.0198 | 0.0441 | 0.0478 | **0.0505*** |
| Sports | HR@5 | 0.0141 | 0.0162 | 0.0154 | 0.0206 | 0.0214 | 0.0217 | 0.0121 | 0.0217 | 0.0267 | **0.0284*** |
| | HR@20 | 0.0323 | 0.0421 | 0.0399 | 0.0497 | 0.0495 | 0.0604 | 0.0344 | 0.0540 | 0.0644 | **0.0656*** |
| | NDCG@5 | 0.0091 | 0.0103 | 0.0114 | 0.0135 | 0.0142 | 0.0143 | 0.0084 | 0.0137 | 0.0177 | **0.0181*** |
| | NDCG@20 | 0.0142 | 0.0186 | 0.0178 | 0.0216 | 0.0220 | 0.0251 | 0.0146 | 0.0227 | 0.0283 | **0.0292*** |
| Toys | HR@5 | 0.0120 | 0.0097 | 0.0166 | 0.0463 | 0.0502 | 0.0274 | 0.0143 | 0.0526 | 0.0571 | **0.0586*** |
| | HR@20 | 0.0312 | 0.0301 | 0.0420 | 0.0941 | 0.0975 | 0.0688 | 0.0235 | 0.1038 | 0.1110 | **0.1148*** |
| | NDCG@5 | 0.0082 | 0.0059 | 0.0107 | 0.0306 | 0.0337 | 0.0174 | 0.0123 | 0.0362 | 0.0392 | **0.0407*** |
| | NDCG@20 | 0.0136 | 0.0116 | 0.0179 | 0.0441 | 0.0471 | 0.0291 | 0.0162 | 0.0506 | 0.0545 | **0.0565*** |
| ML-1M | HR@5 | 0.0467 | 0.1412 | 0.1331 | 0.1444 | 0.1219 | 0.1142 | **0.1579** | 0.1520 | 0.1482 | 0.1557* |
| | HR@20 | 0.1295 | 0.3379 | 0.3187 | 0.3337 | 0.2855 | 0.2921 | 0.3435 | 0.3537 | 0.3431 | **0.3547*** |
| | NDCG@5 | 0.0295 | 0.0916 | 0.0845 | 0.0939 | 0.0798 | 0.0713 | 0.0982 | 0.0969 | 0.0964 | **0.1012*** |
| | NDCG@20 | 0.0524 | 0.1469 | 0.1370 | 0.1476 | 0.1257 | 0.1213 | 0.1510 | 0.1539 | 0.1513 | **0.1570*** |

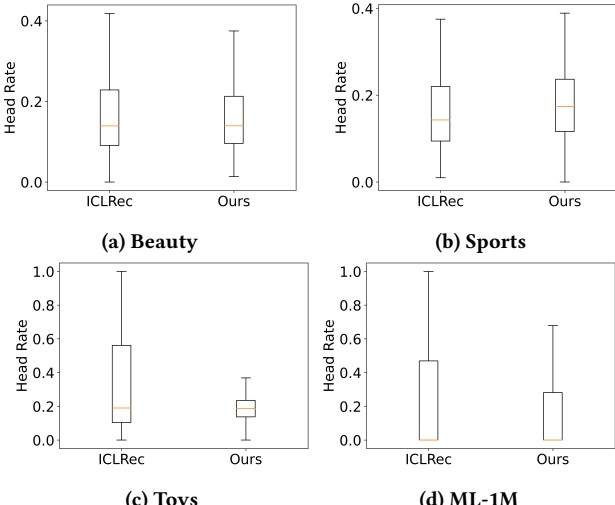

(a) Beauty      (b) Sports

(c) Toys      (d) ML-1M

**Figure 3: The quartile boxplots of head rate.**

the distribution of head rate, defined as the proportion of head sequences within a cluster. As Figure 3 presents, $S^4$Rec has more compact box plots on all four datasets than ICLRec. This observation indicates the distribution of head sequences across clusters generated by $S^4$Rec is more uniform than that of ICLRec. When we eliminate the impact of the head-tail problem on clustering, both the performance on head samples and tail samples increases as shown in Figure 4. Therefore, our adversarial strategy optimizes the clustering distribution, leading to better SR performance.

## 5.4 Ablation Study

Our proposed $S^4$Rec contains two essential modules: cluster-aware two-fold self-distillation (CSD) module and Gradient Reversal (GR)

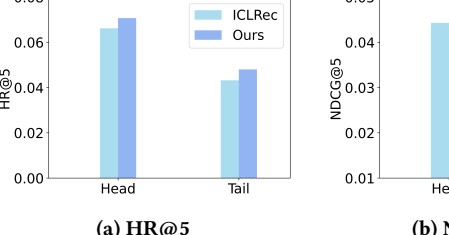

(a) HR@5      (b) NDCG@5

**Figure 4: The head/tail performance of $S^4$Rec compared to ICLRec on the Beauty dataset.**

layer as an adversarial learning module. To understand the impact of the sub-modules of $S^4$Rec, an ablation study is carried out by removing one sub-module at a time. The results reported in Table 3 are based on experiments conducted in the Amazon Beauty dataset. Similar results are also achieved in other datasets.

- SR only: standard sequential recommendation model SASRec
- SR+CSD: apply additional CSD module based on SR
- SR+CSD+GR: apply both CSD and GR module based on SR
- $S^4$Rec w/o CSD: apply GR module and remove CSD on $S^4$Rec
- $S^4$Rec w/o GR: apply CSD module and remove GR on $S^4$Rec
- $S^4$Rec: the complete configuration $S^4$Rec

Overall, both the standard SR model and $S^4$Rec considerably enjoy performance gain from CSD and GR modules. The complete configuration $S^4$Rec is ahead of SR+CSD+GR by 1.61% in terms of NDCG@20. Moreover, the performance of SR and $S^4$Rec both improve consistently as sub-modules are introduced. Based on the results, we validate the effectiveness of the proposed design choices.

**Imapct of Cluster-aware Self-distillation.** We report that the performance declines significantly as the CSD module is removed, regardless of the sequential recommendation model or the proposed $S^4$Rec. Concretely, SR without the help of the CSD module

**Table 3: Ablation study of $S^4$Rec on the Beauty dataset.**

|  | HR@5 | HR@20 | NDCG@5 | NDCG@20 |
|---|---|---|---|---|
| SR only | 0.0374 | 0.0901 | 0.0241 | 0.0387 |
| SR+CSD | 0.0499 | 0.1041 | 0.0332 | 0.0485 |
| SR+CSD+GR | 0.0517 | 0.1044 | 0.0349 | 0.0497 |
| $S^4$Rec w/o CSD | 0.0382 | 0.0922 | 0.0258 | 0.0397 |
| $S^4$Rec w/o GR | 0.0487 | 0.1035 | 0.0329 | 0.0480 |
| $S^4$Rec | **0.0519** | **0.1071** | **0.0348** | **0.0505** |

suffers a 20.21% performance drop at NDCG@20, and compared to the complete configuration of $S^4$Rec, the NDCG@20 of $S^4$Rec without CSD drops 21.39%. We argue that cluster-aware distillation transfers knowledge by extracting additional supervisory signals from the intent-level representation. Therefore, the instance of limited behaviors benefits from intent-level generalization, though the information is limited at the sequence-level. Consequently, the CSD module can efficiently alleviate the problem of insufficient self-supervision signals given users limited behaviors.

**Impact of Head-tail Adversarial Learning.** The adversarial classifier is indispensable since we aim to prevent the semantics of the sequence length from dominating the clustering process. Through the gradient reversal technique, we report that it brings 2.4% and 5.2% NDCG@20 improvement in terms of SR+CSD+GR and $S^4$Rec, respectively. Both models enjoy clear resolution on disentangling head-tail semantics, leading to better performance of intent-level clustering.

## 5.5 Hyper-parameter Sensitivity Study

In this subsection, we investigate and report the impact of a group important hyper-parameters on model performance.

- **Impact of the number of intent cluster $K$.** The number of intent clusters $K$ is vital in the clustering process. We argue that improper choice heavily affects the final performance since $K$ fundamentally influences the user's intention distribution. In terms of HR@5, $S^4$Rec enjoys its best performance at $K = 128$ and suffers mild loss when $K$ is smaller. On the contrary, the performance drops significantly when $K$ is greater than 128. We speculate that sufficiently large $K$ results in false negative representations. Concretely, each intent prototype depletes the resolution of the user's real intentions and further harms the effectiveness of cluster-aware self-distillation.

- **Impact of multi-task objective weight.** As each weight of the objective determines the strength of the gradient during the training process, we vary $\alpha$, $\beta_1$,$\beta_2$, and $\lambda$ at different scales to analyze the impacts on the performance of $S^4$Rec. As shown in Figure 6, we observe that performance reaches the peak with the combination $\alpha$=0.01, $\beta_1$=0.1,$\beta_2$=0.1 and $\lambda$=1.0, and then deteriorates sharply as the weight continues to increase. Coinciding with the discussion of Section 4.3, larger $\lambda$ enforces the CSD module to dispatch the instance into different clusters based on the user's behavior sequence length. As the boxplot Figure 6(b) demonstrates, head users allocated to each cluster concentrate as $\lambda$ increases.

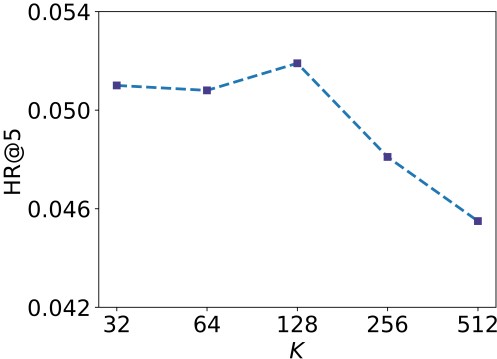

**Figure 5: The impact on the choice of the cluster number $K$.**

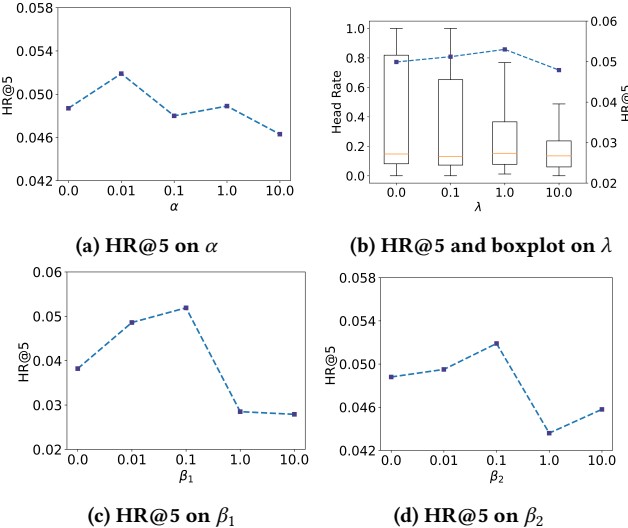

(a) HR@5 on $\alpha$      (b) HR@5 and boxplot on $\lambda$

(c) HR@5 on $\beta_1$      (d) HR@5 on $\beta_2$

**Figure 6: Parameter analysis of $\alpha$, $\lambda$, $\beta_1$ and $\beta_2$ on the Beauty dataset.**

## 6 CONCLUSION

In this paper, we present $S^4$Rec, a practical attempt to address the head-tail problem for the sequential recommendation, which bridges the gap between self-supervised learning and self-distillation methods.

$S^4$Rec utilizes online clustering and adversarial learning modules to optimize user intention clusters unaffected by the sparsity of behavior. Subsequently, the sequence-level contrastive learning considers negative intents augments the representation express capability, while the cluster-aware self-distillation module transfers knowledge from users with extensive behaviors to users with limited behaviors. Experimental results on benchmark datasets confirm and validate the effectiveness of the proposal. We will further investigate how to explore the hyper-parameters automatically used in the model. The code is currently under the legal process and will be soon publicly available.

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
