# OpenReview forum: "Leave No One Behind: Online Self-Supervised Self-distillation for Sequential Recommendation"
_ACM.org/TheWebConf/2024/Conference — TheWebConf24 Oral_

### Official Review · Reviewer_S5s2 · 2023-11-22

**Novelty:** 4
**Technical Quality:** 4

**Review:**

Strengths
Innovative Approach: The S^{4}Rec model innovatively combines self-supervised learning with self-distillation, a novel approach in the field of sequential recommendation systems.
Effective User Grouping: The use of online clustering to group users based on latent intents is an effective strategy to understand user preferences better.
Adversarial Learning Strategy: The inclusion of an adversarial learning strategy to mitigate the bias caused by the length of user behavior data is a significant strength, ensuring a more balanced and fair recommendation process.
Strong Experimental Validation: The model is tested on four real-world datasets, and it consistently outperforms existing methods, indicating its robustness and generalizability.

Weaknesses
Complexity of Implementation: The combination of multiple advanced techniques (online clustering, self-distillation, adversarial learning) might make the model complex to implement and tune.
Potential Scalability Issues: While not explicitly mentioned, the complexity and advanced nature of the model could potentially lead to scalability issues in larger, more diverse datasets.
Dependency on Quality of User Data: The effectiveness of the model is likely highly dependent on the quality and quantity of user interaction data, which might limit its application in scenarios with poor data quality.

The paper addresses a significant problem in sequential recommendation systems and proposes an innovative and effective solution. The combination of self-supervised learning and self-distillation is a novel approach, and the use of adversarial learning to address bias is commendable. The strong performance of the model across multiple datasets further strengthens the paper's contribution. However, the complexity of the model and potential scalability issues are factors that might need further exploration. The success in experimental validation and the novel approach it presents in the field make it a valuable contribution to the literature.

**Questions:**

The paper mentions that traditional clustering methods are time-consuming, have you improved on the traditional methods and have you reduced the time consumption?
Your ablation experiment only demonstrates that adding clustering significantly improves the results of the experiment, but it does not explore in detail how clustering works in training, is it possible to add a set of experiments to illustrate how clustering plays a role in model training? How does the accuracy of clustering change as the epoch increases?

**Reviewer Confidence:**

4: The reviewer is certain that the evaluation is correct and very familiar with the relevant literature

**Scope:**

3: The work is somewhat relevant to the Web and to the track, and is of narrow interest to a sub-community

---

### Official Review · Reviewer_moLd · 2023-11-22

**Novelty:** 4
**Technical Quality:** 4

**Review:**

This paper proposes to combine contrastive learning with self-supervision learning to improve the recommendation effectiveness of cold-start users with limited interactions.

This paper has the following strengths:

+ Combining CL with SSL seems to be an interesting idea for the sequential recommendation.
+ Extensive experiments seem to demonstrate the effectiveness of the proposed method.

However, I have the following concerns.

- Each module of the method, i.e., "online clustering," "adversarial training," and "SSL-based transfer learning," seems ad hoc. Without strong motivation, it seems that each module is loosely combined.
- Since "online clustering" requires clustering the users on the fly, it seems that using offline data passively recorded sequential user behavior may not be sufficient to demonstrate its effectiveness.

**Questions:**

Please refer to the concerns that I listed in my main review.

**Reviewer Confidence:**

3: The reviewer is confident but not certain that the evaluation is correct

**Scope:**

4: The work is relevant to the Web and to the track, and is of broad interest to the community

---

### Official Review · Reviewer_gCdo · 2023-11-23

**Novelty:** 7
**Technical Quality:** 6

**Review:**

The paper introduces a novel cluster-aware approach that improves sequential recommendation performance by adversarially learning concerning both long and short sequences. The problem is well-motivated and the rigorous experiments show remarkable results. Overall, the paper is a good fit for the Web Conference but should be clarified in several aspects.

Strengths:
S1 - The problem is well motivated, e.g., with Figure 1.
S2 - Solid choice of datasets, metrics, and baselines.
S3 - Remarkable and stable results.

Weaknesses:
W1 - Several details need to be clarified.
W2 - Code is not publicly available yet. A location, e.g., a repository, where the code will be made available should at least be provided.

Minor issues:
- Equation 1 misses a | next to L.
- Figure 2 could mark the components with the abbreviations used in the ablation study for cross-checking.
- 5.1.3: BPR is not mentioned as a baseline. Although well-established in the community, a brief introduction would be beneficial.
- 5.1.4: Specify the number of hidden layers and temperature parameters used for the final predictions.
- Figure 3: I recommend fixing the y-axis at 1.0 for (a) and (b) as well.
- 5.4: The statement of "removing sub-module at a time" is not correct. And Typo in "Imapct".
- Conclusion should refrain from referring to the "head-tail problem", but similarly named to the introduction (e.g., sparsity of behavior).

**Questions:**

Q1: In 4.2 prototypes (pi) are not denoted in the Equations, but are shown in Figure 2 in the sequence-level CL. Please clarify.
Q2: What is the formal definition for the head and tail sequences?

**Ethics Review Description:**

-

**Reviewer Confidence:**

3: The reviewer is confident but not certain that the evaluation is correct

**Scope:**

4: The work is relevant to the Web and to the track, and is of broad interest to the community

---

### Official Review · Reviewer_cSkB · 2023-11-24

**Novelty:** 5
**Technical Quality:** 4

**Review:**

The paper introduces a new learning paradigm for sequential recommendation systems: "Online Self-Supervised Self-distillation for Sequential Recommendation (𝑆4Rec)." This approach addresses the challenge of learning users with limited
behavior data. 𝑆4Rec bridges the gap between self-supervised learning and self-distillation methods. It utilizes online clustering and an adversarial learning strategy to group users by latent intents while negating the influence of behavior length. This is followed by a self-distillation process where knowledge is transferred from users with extensive behavior to those with limited behavior. The effectiveness of 𝑆4Rec is demonstrated through experiments on four real-world datasets, showcasing its better performance in sequential recommendation tasks, particularly for users with limited behavioral data.


Pros:
1. The paper's motivation to address the challenge of harnessing self-supervision signals for users with limited behavior data is well-founded. The proposed Head-tail Adversarial Learning approach, designed to mitigate the influence of behavior length, appears to be a thoughtful and rational solution to this issue.
2. The model performance is SOTA is on many metrics.

Cons:
1. Regarding Fig. 1(a) with unlabeled x and y axes, adding clear axis labels and a brief explanation would significantly enhance its interpretability. As it stands, the current illustration does not clearly convey the intended message about the clustering of users with different behavior lengths, as mentioned in the text.
2. Will the distillation from teachers to students affect the full exploitation of users with limited data? Or the sequence category classification?
3. Regarding the ablation study, what are the differences between SR+CSD+GR and 𝑆4Rec?

**Questions:**

See Cons.

**Reviewer Confidence:**

3: The reviewer is confident but not certain that the evaluation is correct

**Scope:**

4: The work is relevant to the Web and to the track, and is of broad interest to the community

---

### Official Review · Reviewer_QVzm · 2023-11-24

**Novelty:** 5
**Technical Quality:** 4

**Review:**

Pros
1.	The paper is generally easy to follow.
2.	They perform visualization on the learned cluster.
3.	The motivation is reasonable.
Cons
1.	The experimented datasets are kind of small, try to use some large-scale datasets, especially some industrial datasets.
2.	It seems the proposed method underperforms some baselines like S^3-Rec, try to beat it or explain it.
3.	The paper writing needs improvement. Write more illustrations in Figure 1 and if possible, do not show single-column figures like Figure 5.
4.	Insufficient discussion on related works or baselines. Some self-supervised sequential recommendation works are ignored, like [1, 2].
5.	The prototype of clustering is similar to this work, I suggest trying to explain your clustering using item group.


[1] Dual contrastive network for sequential recommendation. SIGIR 2022.
[2] Self-Supervised Graph Co-Training for Session-based Recommendation. CIKM 2021.
[3] Mixed Attention Network for Cross-domain Sequential Recommendation. WSDM 2024.

**Questions:**

1.	Could you please add larger datasets?
2.	Could you please explain why the proposed method underperforms some baselines like S^3-Rec?
3.	Please add more baselines like [1,2], or at least discuss them.

[1] Dual contrastive network for sequential recommendation. SIGIR 2022.
[2] Self-Supervised Graph Co-Training for Session-based Recommendation. CIKM 2021.

**Reviewer Confidence:**

4: The reviewer is certain that the evaluation is correct and very familiar with the relevant literature

**Scope:**

4: The work is relevant to the Web and to the track, and is of broad interest to the community

---

### Decision · Program_Chairs · 2024-01-22

**Decision:**

Accept (Oral)

**Comment:**

Scores are ultimately positive, and don't really raise any reasons to reject. While most scores are around the middle, the reviewers seem to explicitly respond positively to the rebuttal and indicate that the issues raised have been adequately addressed.